# Omicron Coronavirus: pH-Dependent Electrostatic Potential and Energy of Association of Spike Protein to ACE2 Receptor

**DOI:** 10.3390/v15081752

**Published:** 2023-08-17

**Authors:** Svetlana H. Hristova, Alexandar M. Zhivkov

**Affiliations:** 1Department of Medical Physics and Biophysics, Medical Faculty, Medical University—Sofia, Zdrave Str. 2, 1431 Sofia, Bulgaria; svhristova@medfac.mu-sofia.bg; 2Institute of Physical Chemistry, Bulgarian Academy of Sciences, Acad. G. Bonchev Str., bl. 11, 1113 Sofia, Bulgaria

**Keywords:** SARS-CoV-2, coronavirus variants, point mutations, spike protein, ACE2 receptor, protein electrostatics, isoelectric point, surface electric potential, folding energy, contagiousness

## Abstract

The association of the S-protein of the SARS-CoV-2 beta coronavirus to ACE2 receptors of the human epithelial cells determines its contagiousness and pathogenicity. We computed the pH-dependent electric potential on the surface of the interacting globular proteins and pH-dependent Gibbs free energy at the association of the wild-type strain and the omicron variant. The calculated isoelectric points of the ACE2 receptor (pI 5.4) and the S-protein in trimeric form (pI 7.3, wild type), (pI 7.8, omicron variant), experimentally verified by isoelectric focusing, show that at pH 6–7, the S1–ACE2 association is conditioned by electrostatic attraction of the oppositely charged receptor and viral protein. The comparison of the local electrostatic potentials of the omicron variant and the wild-type strain shows that the point mutations alter the electrostatic potential in a relatively small area on the surface of the receptor-binding domain (RBD) of the S1 subunit. The appearance of seven charge-changing point mutations in RBD (equivalent to three additional positive charges) leads to a stronger S1–ACE2 association at pH 5.5 (typical for the respiratory tract) and a weaker one at pH 7.4 (characteristic of the blood plasma); this reveals the reason for the higher contagiousness but lower pathogenicity of the omicron variant in comparison to the wild-type strain.

## 1. Introduction

The omicron variant (B.1.1.529, identified in Botswana and South Africa in November 2021, the origin of a pandemic wave of COVID-19 [1,2,3]) of the SARS-CoV-2 β-coronavirus (spherical RNA virus) is characterized by higher contagiousness [4,5] but lower pathogenicity [6,7] as compared with the Wuhan wild-type strain (identified in China in 2019). The infection starts with the association of the virus particles via their S-proteins (a globular homotrimeric glycoprotein integrated in the viral bilayer lipid membrane which envelopes the densely packed RNA chain [8]) to the human angiotensin-converting enzyme (hACE2): a membrane-integrated globular protein involved in the regulation of the blood pressure by inactivation of angiotensin (the strongest vascular constrictor); the viral particles then penetrate into the cells by receptor-mediated endocytosis [9,10]. The ACE2 receptors are located on the human and animal epithelial cells of the alimentary and respiratory tracts, as well as the vascular and excretory systems; the most harmful is infection of the blood vessels, because it causes damage to their epithelial cells and leads to the formation of microthrombi.

The structure of ACE2 macromolecules is different in Homo sapiens and animals, but it is identical everywhere in the human body, only their quantity is different in epithelial cells in various organs. The different clinical pictures of the coronavirus disease during the successive pandemic waves are caused by different structures of the S-protein of the coronavirus variants [11,12]; the difference is determined by more than 30 point mutations in the viral RNA, which lead to substitutions of the corresponding amino acid residues in the S-protein. Some mutations result in a change in the electric charges in the receptor-binding domain (RBD) of the water-soluble S1 subunit by which S-protein associates with the ACE2 receptor. The importance of the electric forces is evident from the experimental fact: the appearance of only one additional positive charge in RBD leads to a 7.5 times higher constant of S1–ACE2 association of the wild-type strain of SARS-CoV-2 in comparison with SARS-CoV-2 (low infecting strain detected in 2003) [13]. This suggests that the increased contagiousness of the newer coronavirus variants is caused by the appearance of additional positive charges on the surface of the S1 subunit.

The importance of the electrostatic interactions in the binding of the S-protein to the ACE2 receptor has been demonstrated in at least two decades of articles [14,15,16,17,18,19,20,21,22,23,24,25,26,27,28,29,30,31,32,33,34,35], but as a rule, they have investigated only RBD, which is a polypeptide containing approximately 200 amino acid residues, instead of the S-protein in its native trimeric form (S-trimer). In particular, a linear correlation of the binding affinity of the RBD to the ACE2 on the pH-dependent net electric charge of RBD of 15 S mutants is established by electrostatic simulations, which allows for predicting the contagiousness of new coronavirus variants [35]. Some stronger RBD–ACE2 binding is found at pH 5.5 more than pH 7.5 by molecular dynamics simulations of the RBD–ACE2 binding, which predicts higher infectivity of both wild-type strain and omicron and delta variants in the upper respiratory tract than in the lung; the binding affinity of the omicron variant is higher in comparison to the delta variant and the wild-type strain [36]. However, it is shown experimentally by surface plasmonic resonance (SPR) that the binding of ACE2 to the S-trimer, S-monomer and RBD is different; after the formation of the ACE2–S-trimer complex, a secondary state in the S-protein emerges, which increases the complex’s stability approximately two hundred times, but this new state appears only when the S-protein is in its trimeric form, it does not occur in the case of the isolated S-monomer and RBD fragment; the authors’ conclusion is that the whole extramembrane domain (including the S1 and S2 subunits) is needed for the formation of a stable complex of the S-protein with the ACE2 receptor [37]. The equilibrium constant of dissociation of ACE2–S-trimer and ACE2–S-monomer complexes measured by SPS shows that the binding of the S-trimer to ACE2 is 60 stronger as compared with the S-monomer [38]. So, the calculation of the binding energy by simulation of only RBD is not the most reliable approach.

There are 15 point mutations in the RBD segment of the polypeptide chain of the omicron variant, 7 of which are charge-changing mutations (in comparison with the wild-type strain) [39,40,41], resulting in the appearance of four NH_3_^+^ groups (N440K, T478K, Q493R, Q498R) and one COO^–^ group (G339D), and the disappearance of one NH_3_^+^ group (K417N) and one COO^–^ group (E484A); this is equivalent by the appearance of three positive net charges.

The purpose of our research was to determine the reason for the different contagiousness and pathogenicity of the omicron variant compared with the wild-type strain (registered in China in 2019). Therefore, employing computer methods, we calculated Gibbs free energy Δ*G*_assoc_ at the association of the S-protein and the ACE2 receptor assuming that Δ*G*_assoc_ is determined by van der Waals and electrostatic forces. The last are pH-dependent because of the ionization of the carboxylic (COOH ↔ COO^–^) and amino (NH_2_ ↔ NH_3_^+^) groups of the amino acid residues. The dissociation constants of the chargeable groups are determined by their chemical structure and the local pH, which is different from pH in the bulk due to the altered concentration of H^+^ and OH^−^ ions caused by the electric field in the vicinity of a given ionized group. In other words, the pH-dependent electrostatic component of Δ*G*_assoc_ at the S1–ACE2 association is determined by the coulomb charges of the two interacting proteins; this requires calculation of the surface electric potential of the S-protein and the ACE2 receptor using atom coordinates in their 3D structures.

There are published pictures of the electrostatic potential on the surface of the S-protein in the literature [42,43,44], but the authors have not taken into account both the pH dependence of the surface potential and the difference between the local and bulk pH. We computed the pH-dependent electrostatic potential and the energy of association of the ACE2 receptor and the S-protein in its trimeric form (S-trimer) using the 3D atomic coordinates of the two protein globules. As a first step, we reconstructed the 3D structure of the S-protein of the omicron variant introducing the point mutations in RBD using the wild-type structure and then calculated the pH dependences of the S–ACE2 association energy, the net charges and the surface electric potential of ACE2 and S-proteins of the wild-type strain and omicron variant. The results reveal that the affinity of the omicron S-trimer to the ACE2 receptor is higher than that of the wild-type strain at pH 5 but lower at pH 7; this finding explains the higher contagiousness but lower pathogenicity of the omicron variant, which are conditioned by the different pH in the upper respiratory tract (pH 5.5–6.5) and blood plasma (pH 7.4).

## 2. Methods

### 2.1. Protein Electrostatics

The surface electric potential of the ACE-2 receptor and S-protein (in three-dimensional form, 3D) of the wild-type strain was computed using the published data in the Protein Data Bank (PDB) structures 7DF3 and 7DF4 [45]. The 3D atomic coordinates of the omicron variant were obtained by introducing charge-changing point mutations (7 in the monomeric and 3 × 7 in the trimeric forms) in the RBD of the S1 subunit using 3D structure of the wild-type strain and employing the method of mutational analyses [46]. The energy of folding Δ*G*_fol_ of the S-trimer and the S–ACE2 complex was calculated assuming that both the trimer (3 × S) and the complex (3 × S + ACE2) are constructed from a single polypeptide chain.

The following programs were used: (a) Site Directed Mutator (SDM) and PBEQ Solver [47,48] for in silico mutagenesis to find the atom coordinates of a mutant model with minimal free energy by analogy with 3D local structures of other native proteins [49,50]; (b) Propka [51,52,53] for protein electrostatics to calculate the surface electrostatic potential and the isoelectric point; (c) Bluues [54] for protein folding energy; (d) Chimera [55] and VMD: Visual molecular dynamics 1.9.2 [56] for visualization of the molecular models and the surface electric potential on the protein globule.

### 2.2. Isoelectric Focusing

A recombinant ACE2 receptor and receptor-binding domain (RBD) of the S-protein of the wild-type strain and the omicron variant were used; the proteins (purity ≥ 95%) were manufactured by Wuhan Fine Biotech Co., Ltd. (Wuhan, China) by synthesizing in embryonic human cells (line HEK 293) after inculcation of corresponding DNA vectors. Lyophilized samples of the proteins were solubilized in triple-distilled H_2_O at a concentration of 0.1 mg/mL. Then, the 150 μL solutions were implanted into overnight-rehydrated IPG strips with a length of 18 cm and a linear pH gradient with ranges pH 3–10 for ACE2 and pH 6–11 for the two RBD, using pI kits according to the instructions of the manufacturer (Cytiva company, Marlborough, MA, USA). The gel electrophoreses were performed by an Ettan IPGphor 3 Isoelectric Focusing System at a constant current of 55 μA per strip in four voltage–time steps: (a) Step and Hold 500 V, 1.0 h; (b) Gradient 1000 V, 1.0 h; (c) Gradient 10 kV, 3.0 h; (d) Step and Hold 10 kV, 30 min. After the electrophoresis, the protein streaks in the IPG strips were fixed in aqueous solution of 10% trichloroacetic acid (1 h), staining in aqueous solution of 25% methanol, 5% acetic acid (30 min) and visualized by coloration with 0.1% colloidal Coomassie Blue G-250 (15 min) (BioRad, Shinagawa, Tokyo). The stained strips were scanned with a GE scanner, and the pI value of proteins was determined with ImageQuant TL 7.0 (GE) software. The isoelectric points pI of the ACE2 receptor and the two RBDs were determined by comparison of the positions of their bands with those of indicative proteins (included in the kit purchased from the producer of IPG strips).

## 3. Results

### 3.1. Molecular Models

Figure 1a shows the complex of the ACE2 receptor with the trimer of the S-protein (3 × S subunits) of the wild-type strain SARS-CoV-2 coronavirus; the molecular model was constructed using the 3D atomic coordinates of the two proteins from the Protein Data Bank (PDB): 7DF4 (ACE2–S-trimer complex) and 7DF3 (S-trimer). The S-protein has two subunits: S1 and S2; the S1 subunit is composed of amino acid residues from 14 to 685 (numbered from the N end of the polypeptide chain). The receptor-binding domain (RBD, 319–541) is located laterally and frontally on the S1 subunit; this suggests that the binding of the S1 subunit to the ACE2 receptor takes up only a part of the surface of the S-trimer.

Figure 1b shows a molecular model of the S-trimer of the omicron variant, which was reconstructed by mutational analysis using the atomic coordinates of the wild-type strain. The model reveals that the charge-changing point mutations in the RBD of the S1 subunit are located on the frontal surface (parallel to the viral lipid membrane) of the S-trimer. This determines the orientation of the S-trimer to the ACE2 receptor (Figure 1): the axis of the complex is perpendicular to the lipid membranes of the viral particle and the epithelial cell to which it is associated.

### 3.2. Association Energy

We determine the Gibbs free energy ΔG_assoc_ at the association of the S-protein (in trimeric form, S-trimer) to the ACE2 receptor (monomeric protein) as a difference of the energy of folding of the ACE2–S-trimer complex and that of its two components (S-trimer and ACE2 monomer):ΔG_assoc_ = (ΔG_fol_)_complex_ − [(ΔG_fol_)_S-trimer_ + (ΔG_fol_)_ACE2_],(1)
where (ΔG_fol_)_S-trimer_ and (ΔG_fol_)_complex_ are calculated assuming that both the S-trimer and the ACE2–S complex are composed as a single polypeptide chain (united 3 × S or 3 × S + ACE2 chains), which is folded in a protein globule; the three chains in the free S-trimer are in a closed conformation (Figure 1c), in the ACE2–S-trimer complex one S-monomer is in an open conformation and remaining two are in a closed one (Figure 1d). The quantities ΔG_fol_ and ΔG_assoc_ have negative value, because the folding of a polypeptide chain and the association of two protein globules are energetically advantageous (Gibbs free energy decreases); the positive value of ΔG_assoc_ means that the protein globules do not associate. The thermodynamic probability of association is equal to the absolute value |ΔG_assoc_| of the free energy decrease at the transition from single-protein macromolecules to their complex.

Figure 2 shows pH dependences of ΔG_assoc_ at the association of the ACE2 receptor to the S-trimer of the wild-type strain or the omicron variant. The greater absolute values of ΔG_assoc_ of the omicron variant in the range under pH 5.7 mean that its association is stronger than that of the wild-type strain. However, above pH 5.7, the ratio of ΔG_assoc_ values is inversed, and then the ACE2–S-trimer association is stronger in the case of the wild-type coronavirus.

To explain the different free energies ΔG_assoc_ at the association of the wild-type strain and the omicron variant, it must be taken into consideration that a few point mutations (which take up too small of an area on the RBD of the S-trimer, see Figure 1b) cannot noticeably change the van der Waals forces between the S-protein and the ACE2 receptor, because their contact surfaces remain almost unchanged. Therefore, the difference in the pH dependences of ΔG_assoc_ can be attributed to the electrostatic forces, which are altered by the seven charge-changed point mutations in the RBD. This requires calculating the net charge and surface electric potential of the S-protein of the wild-type strain and the omicron variant.

### 3.3. Isoelectric Points of 3D Protein Structures

The isoelectric point pI of the 3D models of the ACE2 receptor, the monomeric S-protein, its parts (S1 and S2 subunits), trimeric S-protein (S-trimer) and S-trimer–ACE2 complex were determined by pH dependences *nz* (pH) of the net charge *nz* at *nz* = 0, where *n* is the number of charges (difference between positively and negatively charged groups) and *z* is their valency (*z* = 1 for NH_3_^+^ and COO^−^ groups) (Figure 3). The values of *nz* were calculated by the method of protein electrostatics using literature data of the 3D coordinates of the S-trimer–ACE2 complex of the wild-type coronavirus; the 3D coordinates of the omicron variant were obtained by us by introducing the charge-changing point mutations in the RBD: 7 amino acid residues in the S-monomer and 21 in the trimeric form of the S-protein are replaced; the 3D atomic coordinates of the mutant residues were calculated at a condition of minimum free energy of the protein globules.

Table 1 shows the isoelectric points pI of the S-protein and its subunits, which are caused by substitution of seven charged amino acid residues in the polypeptide chain of the wild-type strain. The charge change is equivalent to the appearance of three additional positive charges in the chain of the omicron variant; this leads to shift ΔpI = [pI_mutant_ − pI_wild_] with 0.5 pH unit in both monomeric and trimeric forms of the S-protein and its parts (RBD, S1 subunit). The independence of ΔpI from the length of the polypeptide chains and the parallel course of the pH dependences of the wild-type strain and the omicron variant in the range of pH 5–9 reveal that the ionization constants of the seven mutant charged groups depends on the neighbor charges in the native 3D protein globule (see Section 4.2).

Comparison of the pI values of the monomeric and trimeric forms of the S-protein shows that their isoelectric points are slightly different, although the three polypeptide chains in the S-trimer are chemically identical. This means that the Coulomb charges of one monomer influence the apparent dissociation constants of the ionizable amino acid residues of the neighbor monomer; the effect is not caused by direct electrostatic interactions, but it is mediated by the different local pH: the local concentrations of H^+^ cations and OH^–^ anions are different from pH in the bulk, and they differ by sign and value on the surface of a protein globule owing to the local electric potential (see Section 3.4 and Section 4.2).

The pH dependence (Figure 3) of the net charge of the S-trimer and the ACE2 receptor shows that the two interacting proteins are oppositely charged at pH 6–7; this means that the electrostatic attraction contributes to the association of the two proteins in the nasal and pulmonary secretions. However, in the blood plasma (pH 7.4), the almost zero net charge of the S-trimer of the wild-type strain predicts that the electrostatic component of the association energy |ΔG_assoc_| is negligible, in contrast to the positively charged S-protein of the omicron variant, which should associate more strongly with the negative ACE2 receptors. However, this inference contradicts the clinical characteristics of the coronavirus disease: the wild-type strain infects vascular epithelial cells more strongly.

The contradiction discloses that the net charge *nz* is not enough to explain the protein interactions at the physiological ionic strength (0.15 mol/L NaCl in blood), because then the electric charges are shielded by the counterions, and the electrostatic forces are short-acting. To reveal the reason, the irregular distribution of the electric charges must be taken into consideration, which forms a nonuniform electric potential on the surface of the S-protein.

### 3.4. Surface Electrostatic Potential

The irregular distribution of the electric charges (determined by the amino acid content and sequence, and 3D structure of the protein globule) determines electrostatic potentials of different signs and values and thereby different electric components of the association energy ΔG_assoc_. The potential on the contact surfaces of the two protein globules is distributed like patches with positive and negative signs, but the predominate potential is negative on the ACE2 receptor and positive on the frontal surface of the RBD domain (Figure 3 and Figure 4, respectively).

Figure 4 discloses that the positive electric potential of the RBD of the wild-type strain is more “concentrated” opposite the patches with negative potential of the ACE2 receptor, but that of the omicron variant is more diffusely distributed. Therefore, the association of the S-trimer to the ACE2 receptor is stronger in the case of the wild-type coronavirus; this inference is in accordance with the clinics of the coronavirus disease: the wild-type strain infects vascular epithelial cells more strongly than the omicron variant.

### 3.5. Isoelectric Points of Unfolded Polypeptide Chains

To verify experimentally the calculated isoelectric point pI (determined from the pH dependences of the net charge *nz* for native 3D structures, Section 3.3), we employed the techniques of isoelectric focusing using recombinant proteins: hACE2 receptor and RBD segments of S-proteins of the wild type and the omicron variant. In this experiment, gel strips with fixed ampholytes (which create a pH gradient) and 8 M urea were used; then, the globular proteins were completely denatured (the polypeptide chains are unfolded). The results (Figure 5) show that the experimental pI values for ACE2 (pI 3.9, and pI 4.1) and RBD (pI 8.8 and pI 9.4 for wild type and omicron, respectively) are lower than the calculated pI 5.4 for ACE2 (Figure 3) and pI 9.2 and pI 9.7 for RBD (Table 1). The obtained pI 9.2 is also somewhat different from the pI 9.02 for RBD measured by capillary electrophoresis [57]. This discrepancy can be caused by: (a) the difference between electrostatic interactions in 3D-folded and unfolded polypeptide chains; (b) the additional electric charges (ionized amino acid residues) of the polypeptide fragments attached to the C end of the used recombinant polypeptide chains (Fc tag for ACE2 and His tag for RBD); and (c) posttranslational modifications which can occur during the syntheses of recombinant proteins in cancer cell culture.

The destruction of the 3D structure of the proteins (unfolding) which emerges in 8 M urea (the medium in the gels used for isoelectric focusing) leads to alteration of the apparent dissociation constant of the chargeable groups of the polypeptide chain, which manifests as a shift of the isoelectric point pI; this effect is explained in Section 4.3. For RBD, the shift ΔpI is approximately 0.3 pH units to lower pH (Figure 6). This value is twice as small as the difference of 0.7 pH units between the experimental pI 8.8 and the calculated pI 9.5 (Figure 5 and Figure 6, respectively) for unfolded RBD of the wild-type strain; for the omicron variant the theoretical–experimental difference (pI 9.8–9.4) is 0.4 pH units, which is close to the ΔpI unfolding effect pI 10.1–8.8. So, the unfolding effect cannot completely explain the observed difference between the theoretical and experimental values of RBD of the recombinant S-protein.

The theoretical unfolding of the ACE2 receptor shifts its isoelectric point with 0.5 pH units to lower pH values, but with 0.3 pH units to higher pH for ACE2-Fc; the last shift is opposite in direction and is four times smaller than the two pH units difference between the calculated pI 6.0 (Figure 7) and the experimental pI 4.0 (the average pI of the two experimental values in Figure 5) for the ACE2+Fc-tag chain (used in the experiment, unfolded in 8 M urea). Therefore, the unfolding effect (increase in pI) is insufficient to explain both quantitatively and qualitatively the lower experimental pI 4 in comparison with the theoretical pI 6.0 for the recombinant ACE2 receptor.

The second reason for the theoretical–experimental differences in pI values is the charge of polypeptide attached to the C end of the recombinant proteins (the so-called tag, used in the techniques of recombinant synthesis of proteins). A His-tag (6 histidine amino acid residues) is attached to the polypeptide chains of the RBD with amino acid sequence 315–535 for the wild-type strain and 319–541 for the omicron variant. This short polypeptide is positively charged at acid pH (due to associated protons) but uncharged at the isoelectric points pI 9.5 and pI 9.8 of the unfolded RBD chains (Figure 6, the wild-type strain and omicron variants, respectively), because the dissociation constant of the histidine’s imidazole groups is in the range pK_a_ 5.6–7.5. This explains why the calculated isoelectric points of the RBD and RBD-His unfolded polypeptide chains coincide.

However, the electric charge of the much longer Fc-tag (18–740) shifts the pI 6.0 of the unfolded ACE2-Fc recombinant polypeptide chain by 0.2 pH units to a higher value in comparison with pI of ACE2 (Figure 7); this increase in pI reveals that the net charge of Fc-tag is positive. The equality of the positive and negative charges is restored at some higher pH due to deprotonation of the charged imidazole groups (Section 4.3); this manifests as pI shift to a higher value, which is caused by attachment of Fc tag to the ACE2 chain. The theoretic pI shift (caused by Fc tag) to a higher value is opposite to the negative difference between the calculated pI 6.0 and the experimental pI 4.0 (Figure 5). This means that the attachment of the Fc tag to the polypeptide chain of the ACE2 receptor cannot explain even qualitatively the discrepancy between the theoretically calculated (Figure 7) and experimental (Figure 5) values of the isoelectric point.

The third factor that can shift the isoelectric point is the carboxylic groups of sialic acid, which is covalently bound to the ACE2 receptor and RBD at their intracellular synthesis; the increased mass of these recombinant proteins with 95% and 33%, respectively, is experimental evidence for the posttranslational modifications (Section 4.5). The additional negative charge shifts pI to lower values, where it is compensated by a decreased negative charge (in the acid pH range) or increased positive charge (in the basic pH range) of the two polypeptide chains. The difference in the compensation is caused by the isoelectric points of the two proteins: pI 4 for the ACE2 receptor (Figure 7) and pI 10 for RBD (Figure 6); in the case of the ACE2 receptor, the dissociation of the carboxylic groups decreases (COO^−^→ COOH), but the proton association to amino groups of RBD increases (NH_2_ → NH_3_^+^). The displacement of the isoelectric point to lower values explains the discrepancy between the calculated (Figure 6 and Figure 7) and the experimental (Figure 5) values of pI.

## 4. Discussion

### 4.1. Energy of Association

The association of two protein globules implies that the Gibbs free energy of the complex is lower than the sum of their individual energies. We calculate the Gibbs free energy of a protein globule as free energy of folding Δ*G*_fol_, which has three components: electrostatic, van der Waals, and hydration. The contribution of the remote Coulomb electrostatic interactions depends on both the concentration of the potential-determining H^+^ ions (in pH ranges where the ionizable groups change their charge) and the ionic strength of the surrounding medium (the indifferent counterions screen the electrostatic potential).

The short-distant van der Waals interactions are determined by the London dispersive, dipole–dipole and dipole-induced dipole forces; the London and the induced-dipole forces are always attractive, and the dipole–dipole forces depend on the mutual orientation of the polarized covalent bonds in the protein globule. The hydrogen bonds and the interactions of hydrophilic amino acid residues with water molecules determine the so-called hydrophobic effect, which leads to folding of the polypeptide chain such that the hydrophobic residues are inserted predominately inside the globule and the hydrophilic—outside; a measure of this effect is the energy of hydration.

As is known, when the S-trimer binds to the ACE2 receptor, one of the three S chains (the one contacting ACE2) undergoes structural changes obtaining so-called open conformation [61,62]; this structural transition is initiated by the binding with the hACE2 receptor [45,63]; it is an order of magnitude slower than the first phase of the binding [37] and can depend on the pH of the medium [64]; the energetic barrier for the reverse transition to close state is much higher [65]. Besides the experiments with isolated S-protein in solution, the experiments with S-protein incorporated in artificial virions also reveal that the three monomers in the free S-trimer are in closed conformation [66]. Acid (negatively charged) aspartate residues, and also histidine in moderately acid medium, participate in the stabilization of the open 3D structure [67].

We take into account the two 3D structural conformations (open and closed) of the S-protein when calculating the free energy of folding Δ*G*_fol_ by using the atomic coordinates of the S-trimer in the ACE2–S-trimer complex (after cutting the ACE2 chain) and those of a free S-trimer; both pdb structures are obtained by the same author by identical cryo-electron microscopy (cryo-EM) techniques [45].

### 4.2. Isoelectric Point and Surface Potential

The different slope of the pH curves of the S-trimer and the ACE2 receptor (Figure 3) reflects the length of their polypeptide chains: ACE2 is a small protein with 12 (negative) charges, but the S-trimer is a large globule with 53 (positive) net charges at pH 6.0. The shift ΔpI of the isoelectric point pI (Table 1) is significant enough to be used as an indicator for charge-changing mutations of the coronavirus.

The calculated values of the isoelectric point pI (Table 1) can be verified experimentally by the techniques of isoelectric focusing. For this purpose, we used gel strips with fixed ampholytes that create a pH gradient: the protein macromolecules stop their electrophoretic motion at a place where the local pH coincides with their pI. In this experiment, the medium contains 8 M urea; this means that the protein is fully denatured: the polypeptide chain is in the conformation of a random coil whose volume is two orders of magnitude bigger than that of the native 3D protein globule; this means that the electrostatic interactions between the neighbor amino acid residues have practically disappeared, an additional cause for this is the high ionic strength which leads to shielding of the Coulomb charges by the counterions. Thus, the pI values calculated by 3D protein electrostatics (Table 1) and the unfolded chains measured by isoelectric focusing can differ.

When the polypeptide chains are unfolded (random coil, full denaturation), the shift ΔpI of the isoelectric point pI (caused by point mutations) does not depend on chain length (respectively, molecular mass), because the charge of the additional chain segments is constant; this is a consequence of the definition of pI as pH at which the negative and positive charges are equal. In this case, the ΔpI reflects the arithmetic summation of the charges, and consequently, the pH-dependent curves *nz* (pH) of the net charge *nz* of the wild-type strain and the mutant protein are parallel. However, when the polypeptide chain is folded (a protein globule with fixed 3D coordinates), subsequent charges can alter the dissociation constants pK_a_ of the chargeable groups and their dissociation degree (–COOH ↔ –COO^−^ and –NH_2_ ↔ –NH_3_^+^), and in this case, the shift ΔpI can depend on the length and 3D conformation of the polypeptide chain. The *nz* (pH) curves of the wild-type strain and the omicron variant are parallel (Figure 3); this reveals that pK_a_ values of the ionizable groups of the seven mutant amino acid residues do not depend on the neighbor charges in the 3D protein globule.

In globular proteins with short polypeptide chain (low molecular mass), the influence of the neighbor charges in the 3D protein globule is less probable; i.e., the dissociation constants of the ionizable groups slightly depend on its 3D folding. Figure 6 and Figure 7 confirm this supposition: the unfolding in the case of a shorter RBD chain (ΔpI 0.3) is less than for the longer ACE2 chain (ΔpI 0.5). This means that the isoelectric point pI of 3D structures (folded) is near the pI of denatured (unfolded) polypeptide chains. This result allows verification of the computer calculations by experimental measurements of the isoelectric point employing the method of isoelectric focusing. However, a problem emerges from the fact that the recombinant proteins bear a polypeptide (attached to the C end of the chain) which shifts the isoelectric point with different values depending on the used tag: the shift is significant for the Fc tag (Figure 7). An even bigger problem arises from the posttranslational modifications that occur when the proteins are synthesized in biological cells; then, the recombinant proteins are glycoproteins which bare sialic acid whose negatively charged groups can significantly shift the isoelectric point pI (Section 4.5). So, in the case of recombinant proteins, the theoretical pI value (calculated by the method of protein electrostatics) is much more reliable than the experimental pI (obtained by the techniques of isoelectric focusing).

### 4.3. Shift of the Isoelectric Point

The changes in the 3D conformation of the polypeptide chain lead to alteration in the distances between the chargeable groups of the amino acid residues, which influences their apparent dissociation constant pK_a_ and degree of dissociation because of alteration in the local pH near to the surface of the protein globule. The calculations by the method of protein electrostatics show a difference in the isoelectric points in folded (3D conformation) and fully unfolded (random coil conformation) polypeptide chains.

The pI values of the RBD chain are shifted with 0.3 pH units to lower values at unfolding in the range of pH 9–10 (where is the isoelectric point, Figure 6). The shift is caused by proton dissociation from the positively charged amino (–NH_3_^+^ ↔ –NH_2_), guanidine (=NH_2_^+^ ↔ =NH ↔) and imidazole (≡NH^+^ ↔ ≡N) groups of the lysine, arginine and histidine amino acid residues, respectively, and the emergence of additional negative charges owing to dissociation of the phenol hydroxyl group of the tyrosine residues; as a result, the positive net charges decrease (the negative charge of the carboxylic groups remains unaltered in the basic pH range where they are fully dissociated).

The unfolding (full destruction of 3D structure) of the polypeptide chain of the ACE2 receptor shifts the apparent pI value to a lower pH with ΔpH 0.46 (Figure 7) because of the same reason: an equation of the local and bulk pH at which equality of the positively and negatively charged groups (COO^−^ = NH_3_^+^) emerges. However, in this case, the alteration of the local pH leads to a shift in the apparent pK_a_ values of the carboxylic groups of aspartic and glutamic acids and the imidazole group of histidine amino acid residues; that leads to increasing of the negative charge, but the positive one remains unaltered, because in the acid pH range, all amino and guanidine groups of the lysine and arginine are fully charged.

### 4.4. Heterogeneity of the Electrostatic Potential

The almost zero net charge of the wild S-trimer (pI 7.3, Figure 3) in the blood plasma (pH 7.4) suggests that formation of the S1–ACE2 complex is not electrostatically conditioned (only van der Waals forces, hydrogen bonds, and hydrophobic interactions can contribute). However, this inference could be true only when the electric charges are evenly distributed on the contacting surface. A specific property of the globular proteins is that the electric charges are irregularly disposed as a result of unique consecution of the amino acid residues in the polypeptide chain and its 3D folding. Because of the irregular disposition of the charges, the electric potential created by them is inhomogeneous: on the surface, there are positive and negative spots; this is better expressed in the isoelectric point (zero net charge) because of the equality of positive and negative charges. Therefore, a positively charged spot on the S-protein (Figure 4) can contact electrostatically with a negative spot on the ACE2 receptor (Figure 3, Insert). Therefore, the electrostatic attractive forces can contribute to the formation of the S1–ACE2 complex even at pH 7.4 (when the S-protein of the wild-type strain is neutral as a whole), together with the non-Coulomb electric forces (London dispersive, charge-induced and permanent dipole–dipole interactions as components of the van der Waals forces), hydrogen bonding, and “hydrophobic” interactions (caused by distortion of the hydrogen bonds between the water molecules located around the hydrophobic amino acid residues), all conditioned by the structurally determined complementarity of the two globular proteins.

The irregularity of the surface electric potential affects most strongly the protein–protein interactions in the absence of electrolytes in the surrounding medium, but its influence is reduced in the blood plasma owing to the shielding effect of the counterions. In the blood plasma (0.15 mol/L NaCl), pulmonary and nasal secretions in the concentration of counterions is not high enough to completely shield the electrostatic potential near the protein surface, it is close to zero only at a large distance from the surface. This means that, at a short distance, the electrostatic interactions are determined by the local surface potential that is created by the disposed irregularly charged groups of the amino acid residues. For example, even at pH coinciding with the isoelectric point, the interaction of the S-protein with the ACE2 receptor is determined by electrostatic forces, together with the van der Waals ones. Under physiological conditions (pH 6–7, 0.15 M NaCl), both types of forces contribute due to the geometric correspondence of the contact surfaces of the two proteins; this determines the specific association of the S-protein to the ACE2 receptor instead of numerous other proteins on the epithelial cell membrane.

### 4.5. Recombinant Proteins and Posttranslational Modifications

In the isoelectric focusing experiment (Figure 5), recombinant RBD and ACE2 receptor proteins are used; this means that a polypeptide (His-tag or Fc-tag) is covalently bound to the C end of their polypeptide chains. The His-tag is a short chain of six histidine amino acid residues whose imidazole groups are uncharged at pH 9–10 where the isoelectric points pI 9.5 and pI 9.8 of RBD are disposed (Figure 6); as a result, the calculated pI values of RBD and RBD-His proteins practically coincide. The Fc-tag is long and positively charged (Section 3.5); its attachment displaces the calculated isoelectric point of the ACE2 receptor to a higher value: from pI 5.8 to 6.0 (Figure 7), on the contrary to the experimentally found shift to lower pI 3.9 and 4.1 of the unfolded ACE2-Fc chain (Figure 5).

The last fact reveals that additional charges (besides the Fc tag) are attached to the ACE2 receptor; it could be caused by posttranslational modifications that emerge in the Golgi apparatus before their excretion from the biological cell used for protein synthesis. The used recombinant RBD-His and ACE2-Fc proteins (Figure 5) are synthesized in human embryonic cells, where the posttranslational modifications can be caused by enzyme (covalent) binding of carbohydrates (or saccharides) to aspartic amino acid residues (N glycosylation). In the case of N-linked oligosaccharides, a polysaccharide dendrite structure occurs, some branches of which end with sialic acid, whose carboxylic group is partially or fully dissociated (the sialic acid is a relatively strong acid with pK_a_ 2.6), respectively, at pH 4 and pH 10 (where the isoelectric points of ACE2 and RBD are disposed); the negative charge of this COO^−^ group causes a shift of pI to lower values.

The posttranslational modifications can be verified by comparison of the calculated and the experimentally determined molecular masses of the recombinant proteins used for the isoelectric focusing (Figure 5). Table 2 shows the contribution of the tag and polysaccharides to the calculated and experimental mass of ACE2 and RBD. The masses of the recombinant proteins ACE2-Fc and RBD-His are calculated according to the amino acid sequences of the ACE2 and the RBD and the polypeptides Fc and His tags used at protein syntheses in cell culture. The masses of the recombinant glycoproteins ACE2-Fc-Gly and RBD-His-Gly, synthesized by expression in human embryonic kidney cells, line HEK 239, are experimentally measured by the manufacturer. The relative increment Δ*M*/*M* of the molecular mass is calculated as a theoretical ratio [(*M*_+tag_/*M*_0_) − 1] of the mass *M*_+tag_ of the recombinant protein (with included tag) to the mass *M*_0_ of the protein, or ratio of the experimental molecular mass *M*_+tag+gly_ (with included tag and polysaccharides) to the theoretical mass with M_+tag_ or without an *M*_0_-included tag: [(*M*_+tag+gly_/*M*_+tag_) − 1] (lower percent) and [(*M*_+tag+gly_/*M*_0_) − 1] [(*M*_+tag+gly_/*M*) − 1] (higher percent).

The data in Table 2 confirm that the experimentally determined masses of the recombinant proteins are higher with 33%, 44% and 28% as compared with those of the “theoretical” ACE2-Fc and RBD-His (whose mass is calculated according the amino acid sequence of the main polypeptide chain and the attached Fc or His tags). These results reveal that the proteins used for experimental measuring of the isoelectric points of ACE2 and RBD (Figure 5) are glycoproteins, as is correctly pointed out by the producer. The negative charge added to the recombinant ACE2-Fc and RBD-His proteins gives the main contribution to the shift of the isoelectric point to lower values, the unfolding effect and the charge of Fc and His tags play a secondary role (Section 3.5).

The literature data corroborate the supposition that the ACE2 and the RBD are glycoproteins, and in the role of sialic acid in the decrease of the saccharoses’ real isoelectric point: (a) it is found that the ACE receptor contains sialic acid and different saccharides: fucose, mannose, galactose, N-acetylglucosamine [68]; (b) the enzyme removing of the bound glucose residues decreases the molecular mass of ACE2 from 120 down to 85 KDa (the last value is near to the mass determined by the amino acid sequence) [69,70]; (c) the higher the mass of the ACE2 receptor, the lower its isoelectic point [71]; (d) the last increases from pI 4.3 to pI 5 after enzyme removing of the sialic acid [72].

### 4.6. Contagiousness and Pathogenicity

The different pH dependences of the free energy ΔG_assoc_ at the association of the S-trimer to the ACE2 receptor (Figure 2), considering the different pH in the respiratory tract [73,74], explain both the facts (known from the epidemiology and clinical characteristics of the coronavirus disease) that the omicron variant has higher contagiousness but lower pathogenicity as compared to the wild-type strain. The higher contagiousness is conditioned by the slightly acid secretions in the upper respiratory tract (pH 5.5–6.5), whose pH is close to the minimum of the ΔG_assoc_ (pH) curve at pH 5.5. However, at the slightly basic pH in the lung (pH 7.6) and the blood plasma (pH 7.4), ΔG_assoc_ of the omicron variant is lower in absolute value; this predicts its weaker association to the ACE2 receptors of the alveoli and the blood vessels and explains its lower pathogenicity, considering that the penetration of the virus particles into the capillaries is easier in the alveoli where their epithelial cells contact immediately with the vascular ones.

The coronavirus evolution [75,76], in which new mutants such as omicron variants XBB and XBB1 have emerged [77], requires investigation of the molecular mechanisms of the transmissibility and pathogenicity. In particular, an important factor for less severe manifestations of COVID-19 disease in the case of mutation D405N of the omicron variant is its impossibility to associate with integrins as possible receptors (different from ACE2) which are capable of binding the S-protein [78].

### 4.7. Comparison with Literature Data

In the investigations of Chinese authors [42,43], the equilibrium constants of dissociation of the RBD–ACE2 complex (defined as molar concentrations at equality of the complex and the two free proteins) are measured by the method of surface plasmon resonance using RDB of the S-protein and human ACE2 receptor; the two experiments are carried out under some different experimental conditions. In the work of Han et al. [42], the measurements are performed in buffer with pH 7.4; then, 50% RDB–ACE2 dissociation appears at 25 nM and 31 nM concentrations of RDB, respectively, for the wild-type strain and the omicron variants. The results of Cui et al. [43] show that the dissociation constant is 68 nM and 24 nM RDB, respectively, for the wild-type strain and the omicron variant; i.e., the association of the omicron RBD to the ACE2 receptor is three times stronger, probably in slightly acid medium (the buffer use is not pointed out, so, at normal partial pressure of CO_2_, the pH is near to pH 6 due to dissociation H_2_CO_3_ ↔ H^+^ + HCO_3_^−^ of the carbonic acid appearing because of the dissolved CO_2_). Comparing the two SPR experiments, it can infer that the omicron S-protein associates with hACE2 stronger at pH 6 than at pH 7.4, contrary to the wild-type strain. These experimental results correspond with our calculations for the association energy of the two S-trimers at pH 5 and pH 7 (Figure 2) and confirm the conclusion that the reason for the different contagiousness and pathogenicity (Section 4.6) of the wild-type strain and the omicron variant is the different (pH-dependent) surface electric potential which is irregularly disposed on the surface of the S-trimer and ACE2 proteins (Section 3.4 and Section 4.4).

In the case of 3D structures pfb7wg6 (pH 7.5) and pfb7wg7 (pH 5.5) used in the article of Cui et al. [43], the free energy for the S-trimer calculated by us is ΔG_fol_ = −262 kJ/mol at pH 5.5 and −204 kJ/mol at pH 7.5; and for the PDB structures (obtained by us by in silico cutting out the RBD segment 319–535 from the same 3D structures of the S-protein), the free energy is ΔG_fol_ = +130 kJ/mol (pH 7.5) and +99 kJ/mol (pH 5.5). Again, the positive ΔG_fol_ values show that the free RBD does not exist as a 3D structure with the same atomic coordinates as being a part of the native S-trimer. By the way, the 3D structures of the S-trimer and RBD of both wild-type and omicron S-proteins used by us have higher absolute ΔG_fol_ values than those of the Chinese authors (all models are obtained by the cryo-EM method), because we have used the most stable 3D structure of the S-trimer as selected by the minimum free energy ΔG_fol_ calculated by us for the numerous models of S-protein available in the Protein Data Bank.

The use of an RBD fragment instead of an S-trimer raises the question for adequacy of its 3D structure to that in the native S-protein. Protein electrostatics calculations for the 3D structure of the omicron variant we used (reconstructed by mutational analyses on the basis of the wild-type S-protein with ID pdb7DF3 in the Protein Data Bank, Section 3.1) reveal that the decrease in the polypeptide chain length reduces the stability of the 3D structures: the electrostatic component of the Gibbs free energy at pH 7.5 is ΔG_fol_ = −924 kJ/mol for the S-trimer, −257 kJ/mol for the S-monomer, −119 kJ/mol for the S1 subunit, and even +38 kJ/mol; the last (positive) value of ΔG_fol_ means that the 3D structure of RBD is thermodynamically instable. This means that the calculation of the surface electrostatic potential for the short RBD chain is incorrect, because the actual 3D atomic coordinates of the protein are different because of the dependence of the degree of dissociation of the ionizable groups on the 3D conformation of the polypeptide chain (Section 4.2). The decrease in the 3D structural stability of the S-protein with reduction of its mass is caused by the decrease in the total van der Waals attraction because of the diminished number of atoms in the polypeptide chain. The found decrease of ΔG_fol_ of the S-protein demonstrates the well-known fact that protein macromolecules are selected large enough in the evolution to ensure their stable functions, although more resources (amino acids and ATP) are spent by the organism for their synthesis.

The comparison of 3D structures pfb7wg6 (pH 7.5) and pfb7wg7 (pH 5.5) (described in [43]) shows the absence of drastic differences in the form of the S-trimer at the two pHs, except of the small loop, which protrudes out from the surface of RBD of one of the three S-monomers. The 3D protein aliment carried out by us shows that the main alterations of 3D coordinates emerge in a short segment, which consist of 28 amino acid residues (466–494, numbered from the N end of the polypeptide chain which is located in the S1 subunit, the C end is in the intramembrane segment of the S2 subunit). Although Cui et al. explain the difference between the two structures with pH-induced changes [43], we suspect that these structural changes are caused by interactions with the solid phase at isolation of the S-trimer fraction by affinity chromatography; in the case that the changes could be indeed pH-induced, they should emerge in the three identical monomers of the S-trimer. This conclusion of ours is corroborated by the experimental cryo-EM structures that the same chain fragment alters its 3D conformation at the association with the hACE2 receptor; according to Xu et al. [45], the S-protein has “closed” and “open” conformations which are different by the hidden and prominent loop of the RBD chain segment (Figure 1c,d), respectively (the loop emerges in the region of the S1–ACE2 binding site); the three subunits of the free S-trimer are in the more stable closed conformation; the binding to the ACE2 allows to overcome the energy barrier and change the 3D conformation to the open state.

The conclusions about the effect of mutations in the omicron variant are made employing different methods and using different protein structures: our results for the binding energy Δ*G*_assoc_ and the isoelectric point pI were obtained by the method of protein electrostatics for the S-trimer [79], but those in the studies of Barroso da Silva et al. [35] and Aksenova et al. [36] by molecular dynamics of RBD. The main difference is that we find out that, at pH 7.4, the omicron variant binds less strongly to ACE2 than the wild-type strain, which we attribute to its lower pathogenicity (Section 3.2 and Section 4.6). Possibly, the difference stems from the fact that we calculated the binding energy of the almost-whole S-protein (except for the three short intramembrane segments) in its trimeric form (S-trimer), whereas in [35,36], only single RBD, which is a small segment (about two hundred amino acid residues) of the polypeptide chain of one of the three S-monomers, was used.

By computer simulations using the molecular dynamics method, in [36] it is found that the binding affinity of RBD to the ACE2 receptor is higher with about 10% for the omicron variant in comparison with the wild-type strain and the delta variant (there is no difference between the last two); for all three variants, the binding is somewhat stronger at pH 5 than at pH 7 due to the higher positive charge of the RBD. This implies that the omicron variant binds more strongly in the upper respiratory tract (pH 5.5–6.5) as compared to the lower one (in the lung, pH 7.6); by this pH-conditioned difference, the authors explain its higher transmissibility and lower pathogenicity. Although this conclusion is reasonable at pH 5, it cannot explain the lower omicron’s pathogenicity, because its binding affinity to hACE2 is higher than those of the wild-type strain and the delta variant also at pH 7; the last means that the omicron variant could easily infect the epithelial cells of the alveoli and the capillaries. However, this inference contradicts the clinics: the pathogenicity of the omicron variant is lower compared to the wild-type strain; this observation could be explained if the binding of the omicron S-protein to the ACE2 receptor is weaker at pH 7.4 (in the blood plasma), taking into consideration that the coronavirus system pathogenicity is a result of damage to the epithelial cells in the blood capillary leading to the formation of microthrombi in the vessels of all visceral organs; the clinics show that the coronavirus disease is most severe in patients with cardiovascular disorders.

Comparing the net charge at pH 7 and the isoelectric point pI of the RBD segment of seven coronavirus variants with literature data for transmissibility, Aksenova et al. [36] have suggested the hypotheses that the positive charge of the RBD increases during coronavirus evolution. This hypothesis was earlier well founded by Barroso da Silva et al. [80] on the basis of 15 coronavirus variants, using more correct electrostatic simulations to calculate a pH-dependent charge of the RBD segment considering the 3D-dependent dissociation constants pK_a_ of the ionizable groups, which determine pI value and the net charge at given pH. Besides the number of the mutants, the two investigations differ in the correctness of the electrostatic calculation: in [36], a simple pI calculator is used which assumes that the electric charges of the ionizable amino acid residues correspond to that in an unfolded (fully denatured) polypeptide chain, ignoring the 3D structure of the protein globule; the pK_a_ values, taken from Stryer Biochemistry (3rd edition), are averaged for the large number of different proteins.

The same pI calculator gives pI 6.33 and 10.8 negative net charges for the S-trimer, while the protein electrostatics program we employed shows pI 7.33 and 11.0 positive net charges at pH 7.0 (Figure 3); the difference of a whole pH unit (10 times different H^+^ concentration at which appears the equality of positive and negative Coulomb charges) and 22 charges compromises the data for pI and the net charge in [36]. The inaccuracy of such pI calculators comes from ignoring the electrostatic interactions between the electric charges in the 3D protein globule, while the protein electrostatics method takes them into account at computing the surface electric potential, which determines the local pH (different from the bulk pH) and, by that, the degree of dissociation of a given chargeable group in a wide pH range around its pK_a_ (Section 4.3). An additional difference comes from the length of the polypeptide chain of the RBD segment: in [36], it is shorter with 12 and 14 amino acid residues (323–531 and 320–528), respectively, for the wild-type strain and for the omicron variant, as compared with that used by us (Table 2).

The electrostatic simulation in [35] takes into account the degree of dissociation of the ionizable groups at given pH on the 3D structure of RBD, which determines their mutual influences by charge–charge coupling. The computed pH dependence of the free energy Δ*G*_assoc_ of RBD–ACE2 associates of 20 coronavirus variants has a deep minimum in the range of pH 5–9, where the Δ*G*_assoc_ has negative values (the positive Δ*G*_assoc_ means impossibility of association); the minimum is deeper with about 15% for the omicron variant in comparison with the wild-type strain (the minimum of Δ*G*_assoc_ means higher binding affinity of RBD to the ACE2). However, in the range of pH 5.5–8.5, the association energy of the coronavirus variants almost does not depend on pH. So, these electrostatic simulations explain the higher transmissibility of the omicron variant but predict its higher pathogenicity, opposite to clinical observations, assuming that these qualities depend on pH 5.5–6.5 in the upper respiratory tract and pH 7.4 in the blood plasma. On the contrary, by protein electrostatics, we calculate the surface electrostatic potential, taking into account the pH-dependent dissociation of the ionizable groups on their 3D location in the protein globule, and the obtained pH dependences of the association energies (Figure 2) explain both the higher transmissibility and less pathogenicity of the omicron coronavirus variant.

## 5. Conclusions

The pH dependences of Gibbs free energy decrease Δ*G*_assoc_ at the association of the S-protein to the ACE2 receptor is different: the absolute value |Δ*G*_assoc_| in the case of the omicron variant is higher in the upper respiratory tract (pH 5.5–6.5) and lower in the lung alveoli (pH 7.6) and in the blood vessels (pH 7.4); this reveals the secret as to why the contagiousness of the omicron variant is higher but its pathogenesis is lower compared to those of the wild-type strain of the SARS-CoV-2 beta coronavirus. The difference in the association energy is due to the alteration of the electrostatic potential on the surface of the S-protein in the region of the receptor-binding domain (RBD), which is caused by seven charge-changing point mutants. The replacement of seven amino acid residues is equivalent to three additional positive charges; it shifts the isoelectric point of the S-trimer with 0.5 pH unit. The pI values determined by isoelectric focusing are significantly less than the calculated pI of ACE2 and RBD; the difference is caused mainly by posttranslational modifications of the glycoproteins ACE2-Fc-Gly and RBD-His-Gly used in the experiment. This demonstrates that the computer calculations with the method of protein electrostatics give more reliable results than the experiment with recombinant proteins.

## Figures and Tables

**Figure 1 viruses-15-01752-f001:**
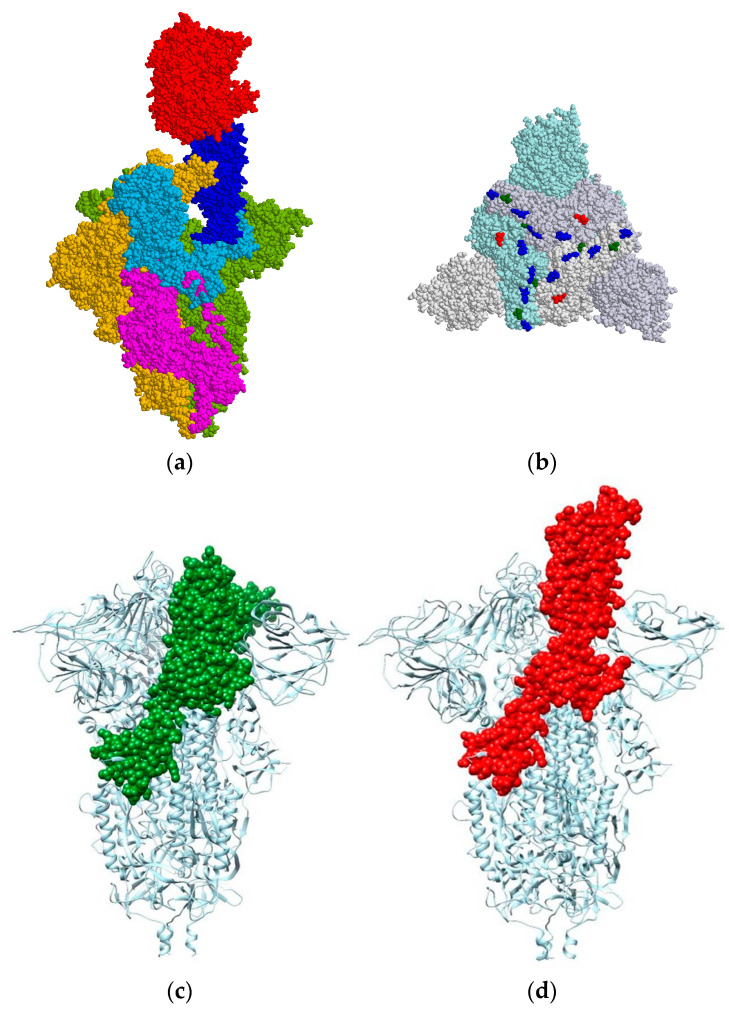
(**a**) Molecular model (lateral surface of the extramembrane domains) of the complex of the hACE2 receptor (the upper red globule) with the S-homotrimer of the SARS-CoV-2 wild-type strain. The segments of one polypeptide chain of the S-trimer are shown in different colors: S1 subunit in blue (14–685 amino acid residues), RBD (segment of S1 chain) in dark blue, and S2 subunit in purple; the second and the third monomers are colored in gold and green, respectively. (**b**) Charge-changing point mutations in the RBD of the S1-trimer of the omicron variant of SARS-CoV-2 (frontal surface). The mutations are colored depending on the charge sign of the new amino acid residues in: blue (the positively charged), green (neutral) and red (negatively charged). (**c**,**d**) Segment (309–685, including RBD) of the polypeptide chain of one S1 subunit in the S-trimer colored according its 3D conformation: closed (green, in free S-trimer) and open (red, in complex with hACE2), respectively, the third and the fourth molecular models.

**Figure 2 viruses-15-01752-f002:**
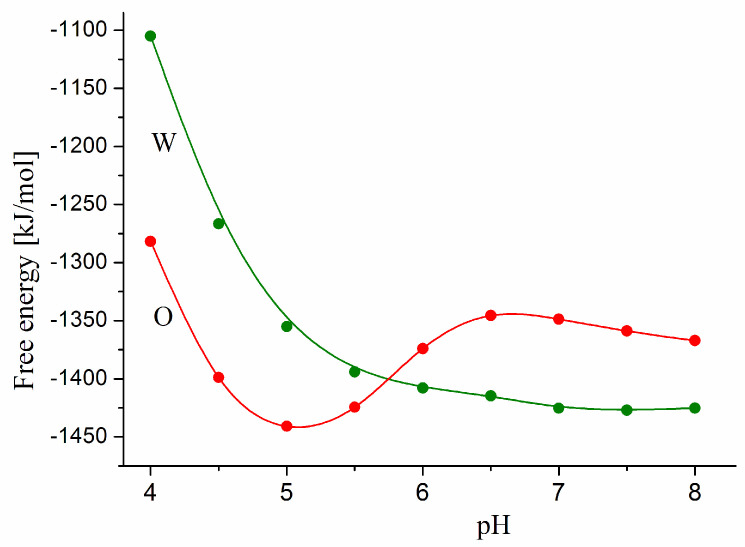
pH dependences of Gibbs free energy at the association of the S-protein (in trimeric form) of the wild-type strain (green curve W) and the omicron variant (red curve O) to the ACE2 receptor of the human epithelial cells receptor.

**Figure 3 viruses-15-01752-f003:**
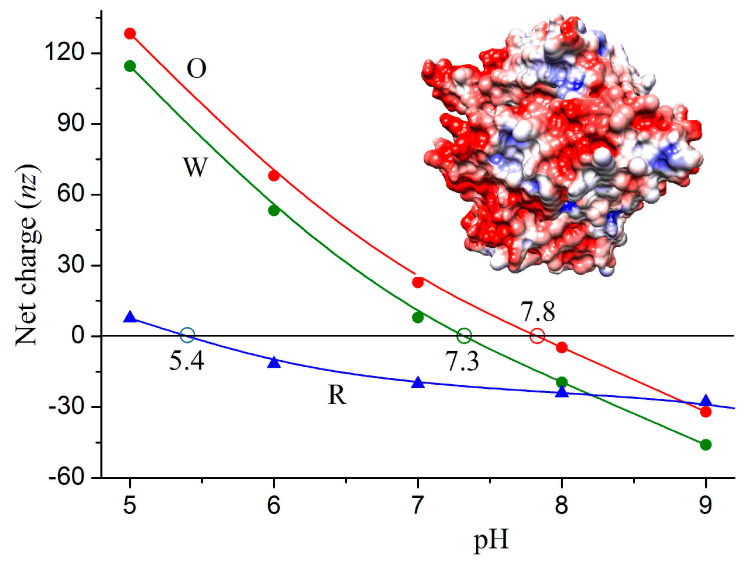
pH dependences of the net charge *nz* of 3D structures (native conformation) of the ACE2 human cell receptor (blue curve R) and trimers of the S-protein of the wild-type strain (green curve W) and the omicron variant (red curve O) of the SARS-CoV-2 coronavirus. *Insert*: Virus-binding surface of the ACE2 protein whose electric potential at pH 6.5 is colored in red (negative) and blue (positive).

**Figure 4 viruses-15-01752-f004:**
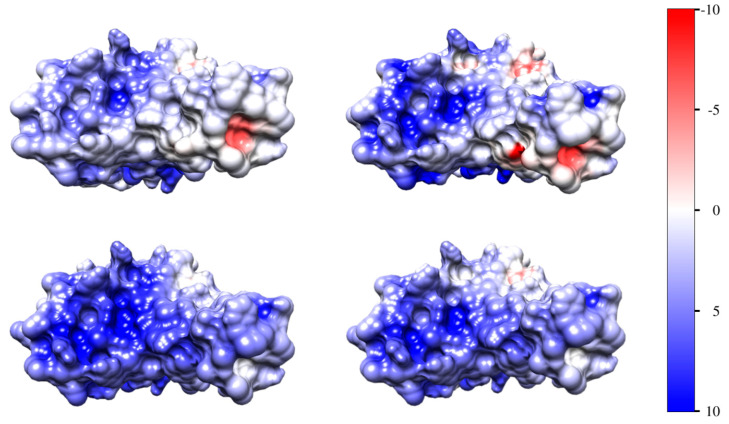
Surface electrostatic potential on the surface of the RBD domain of the wild-type strain (upper two pictures) and the omicron variant (the lower two ones) at pH 5.5 (on the left) and pH 7.4 (on the right). The protein globules are oriented with the surface which contacts with the ACE2 receptor. The electric potential φ = *kT*/*e* is calculated and visualized by color (blue—positive potential, red—negative) in the range ±10 kT/e units, where *k*—Boltzmann constant (J/K), *T*—absolute temperature (°K) and *e*—elementary (proton) charge (C)]; 1 *kT*/*e* (J/C) = 26.7 mV at 37 °C.

**Figure 5 viruses-15-01752-f005:**
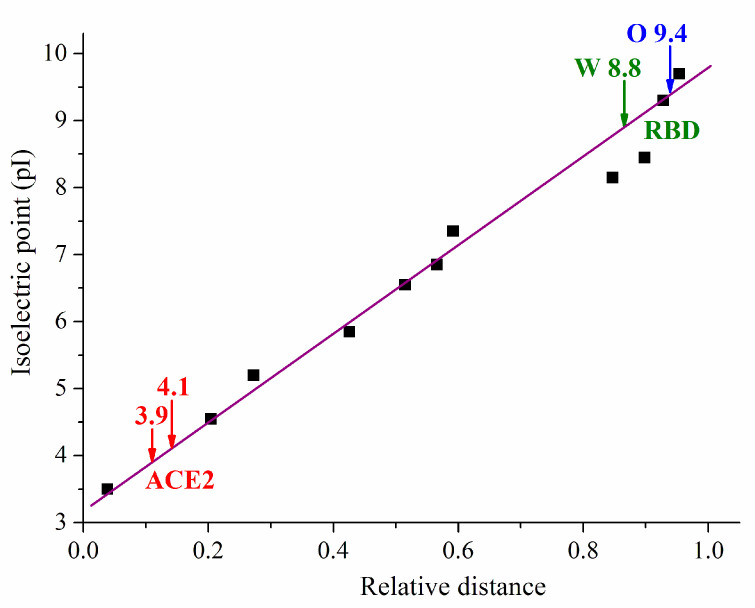
Dependence of band position of indicative proteins (black squares), ACE2 receptor (red arrows 3.9 and 4.1, respectively at pH 3.9 and pH 4.1), RBD of the S-proteins of the wild-type strain (green arrow W 8.8 at pH 8.8) and the omicron variant (blue arrow O 9.4 at pH 9.4) on the relative (dimensionless) distance in a gel with fixed ampholytes. The violet line corresponds to the local pH, whose values were measured by microelectrodes in the same type of gel strip by the producer [58]. The distances from the track start are divided by the strip length (both measured in millimeters). The indicative proteins (the black squares) are: amyloglucosidase (pI 3.5); soybean trypsin inhibitor (pI 4.55); β-lactoglobulin A (pI 5.2); bovine carbonic anhydrase B pI (5.85); human carbonic anhydrase B (pI 6.55); horse myoglobin (acetic band, pI 6.85); horse myoglobin (basic band, pI 7.35); lentil lectin–acidic band (pI 8.15), lentil lectin–basic band (pI 8.65); trypsinogen (pI 9.30) and cytochrome *c* (pI 9.44 [59,60]).

**Figure 6 viruses-15-01752-f006:**
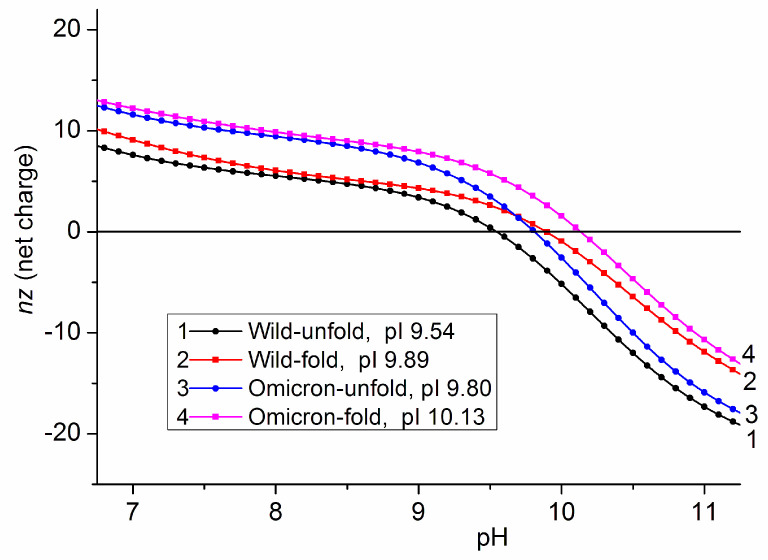
pH dependences of the net charge *nz* of unfolded (curves 1, 3) and 3D folded (curves 2, 4) polypeptide chain of RBD of the S-protein of the wild-type strain (curves 1, 2) and omicron variant (curves 3, 4) of the SARS-CoV-2 coronavirus. The amino acid sequences of the two polypeptide chains correspond to those of the recombinant proteins used in the isoelectric focusing experiment: His tag is attached to the C end of the polypeptide chains. The *nz* values are calculated with step 0.1 pH units; the isoelectric point pI is equal to pH at *nz* = 0.

**Figure 7 viruses-15-01752-f007:**
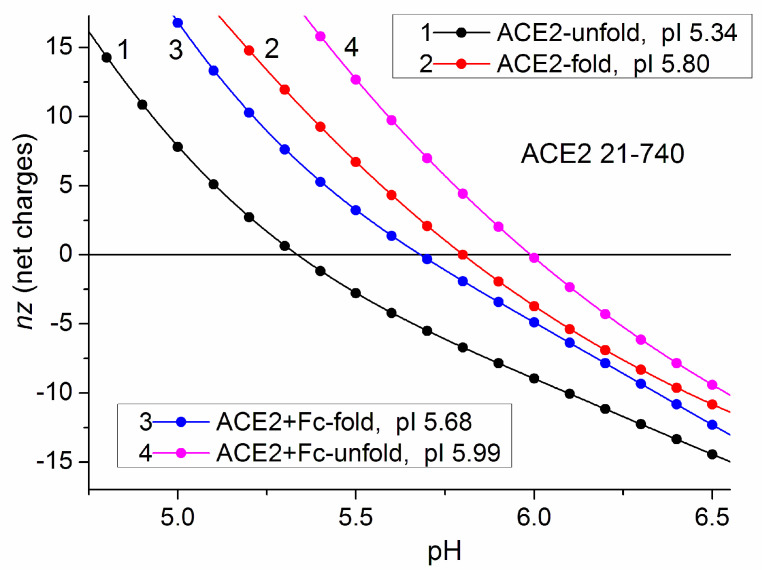
pH dependences of the net charge *nz* of the ACE2 receptor (curves 1, 2) and ACE2 with attached Fc-tag (curves 3, 4) in 3D-folded (curves 1, 3) and -unfolded (curves 2, 4) conformation of the two polypeptide chains. The amino acid sequences of the two polypeptide chains correspond to those of the recombinant proteins used in the isoelectric focusing experiment. The Fc-tag is attached to the C end of the polypeptide chains. The *nz* values are calculated with step 0.1 pH units; the isoelectric point pI is equal to pH at *nz* = 0.

**Table 1 viruses-15-01752-t001:** Isoelectric points of 3D models of the wild-type strain and the omicron variant.

Model	Wild	Omicron
RBD monomer	pI 9.2	pI 9.7
S1-monomer	pI 8.7	pI 9.2
S2-monomer	pI 6.9	pI 6.9
S-monomer	pI 7.4	pI 8.0
S-trimer	pI 7.3	pI 7.79
S-trimer–ACE2	pI 6.5	pI 7.0

**Table 2 viruses-15-01752-t002:** Theoretical and experimental molecular masses of the human angiotensin-converting enzyme (ACE2 receptor protein) and the receptor-binding domain (RBD) of the S-protein of the wild-type strain and the omicron variants of the SARS-CoV-2 beta coronavirus.

Protein	Expression Region	Amino Acid Residues	Molecular Mass [kDa]	Mass Increment Δ*M*/*M*
ACE2	–	723	83.68	–
ACE2-Fc	18–740	952	109.4	31%
ACE2-Fc-Gly	18–740	952	145	33%; 73%
wild RBD	315–529	221	24.85	–
wild RBD-His	315–535	227	25.68	3%
wild RBD-His-Gly	315–535	227	37	44%; 49%
omicron RBD	319–535	223	25.03	–
omicron RBD-His	319–541	229	25.86	3%
omicron RBD-His-Gly	319–541	229	33	28%; 32%

## Data Availability

Not applicable.

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
