# Peer review of "Omicron Coronavirus: pH-Dependent Electrostatic Potential and Energy of Association of Spike Protein to ACE2 Receptor"

_viruses, 2023, doi:10.3390/v15081752_

Round 1
Reviewer 1 Report
Abstract: The abstract provides a general overview of the study's objective and findings, but it lacks clarity and conciseness. It could benefit from a more precise summary of the research question, methods used, and key results. Additionally, the abstract should mention the limitations of the study to provide a balanced perspective.
Introduction:
- The introduction provides background information on the omicron variant of SARS-CoV-2 and its association with ACE2 receptors. However, it lacks proper citations for some statements, making it difficult for readers to verify the claims made.
- The section could benefit from a clearer structure and organization to present the information in a more coherent manner.
Methods:
- The methods section lacks sufficient detail to allow replication of the study. It does not mention specific software versions or parameter settings used in the calculations.
- The description of the experimental techniques used, such as isoelectric focusing, is incomplete and lacks information about controls and potential sources of error.
- References to previous studies or protocols for the methods employed are missing, making it challenging for readers to understand the methodology in its entirety.
Discussion:
- The discussion section does not provide a comprehensive analysis and interpretation of the obtained results. It mainly focuses on describing the energy of association without discussing the broader implications or biological significance.
- The section lacks critical analysis and comparison with existing literature. It would benefit from a discussion of how the findings align or differ from previous studies, helping to contextualize the results.
- The limitations of the study, including any potential sources of bias or uncertainty, are not mentioned, reducing the robustness of the conclusions drawn.
Overall, the manuscript requires significant revisions to address the shortcomings mentioned above. Improvements should be made in terms of clarity, organization, methodological detail, critical analysis, and contextualization of the findings. Additionally, the manuscript should include discussions on limitations and potential future research directions to provide a more comprehensive and balanced perspective.
The English grammar and language in the manuscript have some issues that need attention. Here are some examples:
-
Abstract: "We compute pH-dependent electric potential on the surface" - The verb "compute" should be in the present tense, so it should be "We compute the pH-dependent electric potential..."
-
Abstract: "The calculated isoelectric points of ACE2 receptor" - The word "calculated" should be followed by "the" to make it grammatically correct: "The calculated isoelectric points of the ACE2 receptor..."
-
Abstract: "experimentally verified by electric focusing" - It seems that "electric focusing" is a typo and should be "isoelectric focusing."
-
Introduction: "The omicron variant (B.1.1.529, identified in Botswana and South Africa in November 2021, the origin of a pandemic wave of COVID-19" - The sentence is incomplete and lacks proper punctuation.
-
Introduction: "The infection starts with the association of the virus particles via their S-proteins (membraneintegrated globular homotrimeric glycoprotein" - The word "membraneintegrated" is missing a space between "membrane" and "integrated."
-
Methods: "The following programs were used for molecular models, protein electrostatics and docking: PHEMTO [xx], Propka [xxi, xxii, xxiii], Bluues [xxiv], PBEQ Solver [xxv, xxvi], Chimera [xxvii], VMD: Visual molecular dynamics 1.9.2 [xxviii] and SDM – Site Directed Mutator [xxix]." - The list of programs could be formatted more clearly by using bullet points or separating each program into a separate sentence for better readability.
-
Discussion: "The fact of association of two protein globules means that the Gibbs free energy of the complex is lower than the sum of their individual energies." - The phrase "the fact of association" could be rephrased to improve clarity: "The association of two protein globules implies that the Gibbs free energy of the complex is lower than the sum of their individual energies."
These are just a few examples, but there may be other instances throughout the manuscript where language and grammar can be improved. It is important to thoroughly proofread the manuscript to ensure clarity and correctness.
Author Response
We are very thankful for this review. All specific instructions are very helpful and have been implemented (corrections are colored blue). Of course, I shall read the proofreading very carefully to ensure clarity and correctness of the text. But I am not in a position to completely revise the text according to the general instructions given in the first part of the review, although I have tried to follow them in the revision. Obviously, the reviewer is a very experienced author, I presume of articles in biology, where the description of many details in the experiment and interpretation is requested. My modest experience is in the field of physical and biophysical chemistry, where brevity and clarity are valued, as I have sought to achieve in writing the present manuscript, but so as to make the article accessible to a wider readership. Observing how my PhD-students read articles, I saw that they mainly pay attention to the results, so I divided the interpretation into two frequencies: the main ones in Results and Interpretation, and the more in-depth ones in Discussion. This will make it easier for young researchers who are of the digital generation to read the article.
Reviewer 2 Report
Authors computed the pH-dependent electric potential on the surface of the interacting globular proteins, as well as the pH-dependent Gibbs free energy during the association of the wild-type strain and the omicron variant. They found that the association for omicron variant is higher in the respiratory tract and lower in the blood vessels, supporting the reason why the contagiousness of the omicron variant is higher but its pathogenesis is lower when compared to these of the wild type strain.
Overall the manuscript is well done and well written. I have only few minor points:
- References must be updated, I found the lack of several recent manuscripts focused on the dangerousness of the most recent omicron variants (see for instance Scarpa et al., 2023; Ciccozzi et al., 2023; Cao et al., 2023; Tosta et al., 2023)
- In the text there are several typos that should be fixed. Please check it well.
After fixing these points in my opinion the manuscript can be published.
Author Response
We are very thankful for the indicated citations, they have been added in Discussion (the second paragraph of Section 4.6), which will improve the article. The typos have been corrected, the changes are colored blue.
Reviewer 3 Report
Recently, an article was published on this topic, which is not considered by the authors of the article: “The Increased Amyloidogenicity of Spike RBD and pH-Dependent Binding to ACE2 May Contribute to the Transmissibility and Pathogenic Properties of SARS-CoV-2 Omicron as Suggested by In Silico Study”. Int J Mol Sci. 2022 Nov 4;23(21):13502. doi: 10.3390/ijms232113502.
Therefore, the article should be rewritten taking into account the obtained results. The article is written carelessly, there are many mistakes. Some of them are listed below.
Comments:
References to articles must be formatted in accordance with the journal's requirements.
Abstract:
In the phrase "leads to stronger S1-ACE2 association at pH 5.5 (in the respiratory tract) and weaker one at pH 7.4 (the blood plasma)", it would be clearer to specify that pH 5.5 is typical for the respiratory tract and pH 7.4 is typical for blood plasma, rather than implying that these conditions exist exclusively in these locations.
Methods:
The name of the cell line "HEK 239" might be a typographical error. It's generally known as "HEK 293". Please, check.
Results:
In Figure 1 description, "recidues" should be "residues", "different colored" should be "differently colored", "omicon" should be "omicron", "mutatinns" should be "mutations", "collored" should be "colored", "aminoacid" should be "amino acid".
Page 3. The phrase "the molecular model was constructed using 3D atomic coordinates of the two protein in Protein data bank" could be better phrased as "the molecular model was constructed using the 3D atomic coordinates of the two proteins from the Protein Data Bank".
Page 12. Please, check the description "the carboxylic groups of asparagine and glutamine acids". Maybe, the amino acids are aspartic acid and glutamic acid.
Page 12. The phrase "a positively charged spot on the S-protein (Figure 4) can contact electrostatically with a negative spot on the ACE2 receptor (Figure 3, Insert)" implies an over-simplification. The interaction between proteins is not solely dictated by opposite charges but also involves structural complementarity, hydrogen bonding, and hydrophobic interactions.
Page 13. The phrase "posttranslational modifications can be caused by enzyme (covalent) bounding of carbohydrates (or saccharide)to asparagine aminoacid residues (N- glycosylation), which leads to forming of polysaccharide dendrite structure" is not entirely accurate. Glycosylation doesn't lead to a "dendritic structure", but rather to the addition of carbohydrate groups. If by "dendritic" the authors mean branching, it's important to note that this only occurs with certain types of glycans (i.e., N-linked oligosaccharides), and not all glycans are branched. Please, check.
Table 2. "receptor-biding" should be "receptor-binding".
Page 14.
"aminoacid" should be corrected to "amino acid", "remuving" should be "removing", "secvention" should be "sequence", "increses" should be "increases", "saccharoses" should be "saccharides".
Please check throughout the document, "aminoacid" should be written as "amino acid".
Poorly designed, many mistakes
Author Response
We are very thankful for the very accurate review of the manuscript. Although the typos would have been corrected by the language editor, we found the substantive comments very helpful. All suggested modifications have been accepted (modifications are given in blue), only "aminoacid" has not been replaced by "amino acid", as both spellings occur in the literature, we prefer the former.
We are especially grateful for the indicated article by Aksenova at al. We did not notice it, as our article was written a year ago (only the isoelectric focusing experiment is added in the current one), when their article was not published. We are satisfied that both research groups have independently arrived at an explanation for the higher transmissibility but lower pathogenicity of the omicron variant, this is particularly valuable as the results were obtained with different methods. In the Discussion, a section "4.7. Comparison with literature data" has been added that compares the two studies, reporting some differences in the approaches and the results.
Round 2
Reviewer 1 Report
The manuscript is now in better form for publication.
Author Response
Thank you again for your review, it really helped us to improve the text.
Reviewer 3 Report
It should be noted that Cui et al., 2022 (10.1016/J.CELL.2022.01.019) showed pH-dependent structural changes in the S-protein, they affect the RBD domain. In Figure S2, they show the structures at different pH. It would be nice to discuss these data. How does this affect the electrostatic potential?
Electrostatic RBD surfaces of different variants are shown in Figure 4I and Figure S6 in Han et al., 2022 Cell (10.1016/j.cell.2022.01.001), showing the difference between the variants. It seems to me that the authors could discuss these data - how everything correlates.
It is not clear why the authors expect ACE2 binding to the whole S-protein, and besides, they consider this to be more correct.
The boundaries of the RBD domain are generally established, RBM is much smaller than RBD, if the authors considered this to be normal using extended RBD, but there is no reason to consider this more correct than in the article (Int J Mol Sci. 2022 Nov 4;23(21):13502. doi: 10.3390/ijms232113502), where the calculation was made for RBD and for RBM.
This article (Int J Mol Sci. 2022 Nov 4;23(21):13502. doi: 10.3390/ijms232113502) was submitted on August 8 (the date is in the article), and the authors' preprint was only on the 26th August. This article takes precedence in any case, and I think it would be correct to cite it first time in the Introduction, and not at the very end of the work.
Author Response
Thank you very much for your second review. It has allowed us to clear up important questions about the stability (pH-dependant and chain length) of the S-protein. The new text (coloured in blue) is added in Introduction (the last paragraph) and mainly in Discussion (the first four paragraphs in Section 4.7).
We are especially thankful for pointing out the two articles of Chinese authors (published in 2022, after the first version of our article had been written and published as preprint, and because of that missed by us); the experimental data on the RBD-ACE2 dissociation (obtained by surface plasmon resonance) are very important and useful for us because they confirm our calculations.
Round 3
Reviewer 3 Report
I have the following comments:
1) Both Cui et al and Han et al. were published in February-March 2022, half a year before the authors sent their preprint.
So, the answer of the authors looks ridiculous. They just didn't read literature.
2) In the work of Cui et al., it is simply not written whether they used a buffer or not. The protein was removed from the column in 20 mM Tris buffer pH 7.5 and 200 mM NaCl
So, with a high probability, a buffer close to this was also used in the experiment. Why did the authors decide that they did SPR without a buffer?
It is pointless to make assumptions about what kind of buffer the Chinese used, the authors can write to them and ask.
3) Directly comparing the binding results of Wuhan RBD and Omicron in two different studies, as the authors do in paragraph 4.7, is not the point.
Especially like this: "I.e. the omicron S-protein associates with hACE2 stronger at pH 6 than at pH 7.4."
From different studies, an assumption can be made, which is then experimentally verified in one study.
The Chinese didn't suggest that pH is related to RBD binding to ACE2, don't attribute words to them that they didn't write and guess for them.
The authors of paper [20] were the first to put forward such an idea in the article, and this should be directly displayed.
In general, it is better for the authors to rewrite this whole piece more adequately.
4) The reasoning that a whole S-protein is better than RBD is also unsubstantiated. Structurally, this is a separate domain, especially in an open conformation; the surface is known to interact with ACE2. Part of the S-protein is generally cleaved off upon contact with ACE2, very strong structural rearrangements occur there, but not in the RBD region. Do the authors take these structural changes into their calculations?
Then let the data then be presented for pre-fusion and post-fusion conformations.
This one is bad too: "in the case that the changes could indeed be pH-induced, they should emerges in the three monomers which are identical by their amino acid content, sequence and 3D structure."
These three protomers are in different conformations - open and closed. Changes are visible in the open conformation, which occurs in the closed conformation is difficult to register, there are more contacts with other parts of the S-protein and they are not required to behave in the same way. By the way, this is discussed in [20].
In general, I would like the authors to remove the phrase that paper [20] was published two months after preprint [48]. This preprint was posted 26 August 2022, paper [20] was submitted 8 August 2022.
There are many errors.
Author Response
Dear Reviewer,
Thank you very much for your three critical reviews which stimulated us to reduce our flagrant incompetence. In the third revision of the manuscript 31 references, two pictures (Figure 1c,d) and seven paragraphs (one in Introduction, two in section 4.1 and four in section 4.7 of Discussion, all coloured in blue) are added.
1) Both Cui et al and Han et al. were published in February-March 2022, half a year before the authors sent their preprint. So, the answer of the authors looks ridiculous. They just didn't read literature.
AZ: Yes, indeed. The reason is that our article was basically written a year before the preprint (in summer of 2021), but was not finished due to our busyness with a large anticancer project, as opposed to the small corona-virus project for young scientists that Dr. Hristova is the leader of. Of course, this does not excuse us. But the literature search is hampered by the fact that there is a huge amount of published work on corona virus. Perhaps for this reason, the article by Aksenova at al. omitted to cite the work of Barroso da Silva at al. (published in J. Physical Chemistry B, one of the strongest physicochemical journals of the American Chemical Society), where a linear correlation of RBD-ACE2 binding affinity from the net RBD charge of corona-virus mutants was found by electrostatic simulations.
2) In the work of Cui et al., it is simply not written whether they used a buffer or not. The protein was removed from the column in 20 mM Tris buffer pH 7.5 and 200 mM NaCl. So, with a high probability, a buffer close to this was also used in the experiment. Why did the authors decide that they did SPR without a buffer? It is pointless to make assumptions about what kind of buffer the Chinese used, the authors can write to them and ask.
AZ: If the Chinese authors had used buffer, they would have written what kind it was and what pH (this is mandatory when describing experimental results). In their experiments they used two types of buffers, but just because the protein was isolated at a certain pH does not mean that the SPR experiment was conducted at the same pH. It is likely that the isolated RBD solution was dialyzed to remove NaCl, as its high concentration did not allow the SPR experiment to be conducted qualitatively (at an ionic strength of 200 mM the electrostatic attraction between RBD and ACE2 was weak due to charge shielding). To remove this doubt, we queried the authors by e-mail on 25 July, but have had no reply so far.
3) Directly comparing the binding results of Wuhan RBD and Omicron in two different studies, as the authors do in paragraph 4.7, is not the point. Especially like this: "I.e. the omicron S-protein associates with hACE2 stronger at pH 6 than at pH 7.4." From different studies, an assumption can be made, which is then experimentally verified in one study.
AZ: The same method was used in the studies of the two Chinese collectives, so the results should match. If Cui et al. had used a pH 7.4 buffer, their results would be inconsistent with that of Han at al. This is another (indirect) indication that Cui et al. did not use a buffer. The reviewer's assertion that a comparison can only be made if the results of two studies are verified in a third experiment is the opinion of a computer specialist who has no idea of the difficulties that arise, especially with the complicated SPR technique. In practice, comparing results from different authors is unavoidable.
The Chinese didn't suggest that pH is related to RBD binding to ACE2, don't attribute words to them that they didn't write and guess for them. The authors of paper [20] were the first to put forward such an idea in the article, and this should be directly displayed. In general, it is better for the authors to rewrite this whole piece more adequately.
AZ: This sentence is corrected (the first paragraph in Section 4.7). There is nothing new in the statement that the binding of two proteins is pH-dependent, since the electrostatic component depends on the ionization of the ionizable groups of the amino acid residues. In the particular case of the S-protein and the ACE2 receptor, this claim has been substantiated in two dozen articles, some of which are cited in the third (current) revision of our manuscript (the third paragraph of Introduction).
4) The reasoning that a whole S-protein is better than RBD is also unsubstantiated. Structurally, this is a separate domain, especially in an open conformation; the surface is known to interact with ACE2. Part of the S-protein is generally cleaved off upon contact with ACE2, very strong structural rearrangements occur there, but not in the RBD region. Do the authors take these structural changes into their calculations? Then let the data then be presented for pre-fusion and post-fusion conformations.
AZ: The binding affinity to hACE2 is different for RBD-trimer and RBD-monomer (see the third paragraph in Introduction). We have performed our calculations with both conformations (closed in all three monomers in the free S-trimer and open in one of the trimers in the S-trimer-ACE2 complex) (see the fourth paragraph in section 4.1), so we account for the structural changes. For clarity, we have prepared two new figures (Figure 1c,d). The reviewer's statement "Part of the S-protein is generally cleaved off upon contact with ACE2" is incorrect: this cleavage occurs not at once, but at a later stage by a proteolytic enzyme in the endosomes/lysosomes (when the viral body has already been absorbed by the epithelial cell by phagocytoses; in the literature two additional mechanism are speculatively suggested: by membrane enzyme before the phagocytoses or even at intracell syntheses of S-proteins of the new viral particles. The terms “pre-fusion and post-fusion” concern the S-protein states before and after fusion of two lipid membranes (the viral and epithelial cell), but the using of these terms is inappropriate in the case of protein solutions without viral particles and biological cells; therefore we use “closed and open state/conformation”. In any case, the “post-fusion conformation” does not concern our investigation.
This one is bad too: "in the case that the changes could indeed be pH-induced, they should emerges in the three monomers which are identical by their amino acid content, sequence and 3D structure." These three protomers are in different conformations - open and closed. Changes are visible in the open conformation, which occurs in the closed conformation is difficult to register, there are more contacts with other parts of the S-protein and they are not required to behave in the same way. By the way, this is discussed in [20].
AZ: The transition from the energetically more advantageous closed to the less advantageous open conformation occurs upon binding of S-protein to ACE2 receptor, then the energetic barrier of the structural transition is overcome (see the third paragraph in section 4.1). The changes in 3D structure affect a insignificant portion of the chain of one of the three S-monomers (colored portions in Figure 1c,d). If pH induces such a transition, it should be observed at all three monomers.
In general, I would like the authors to remove the phrase that paper [20] was published two months after preprint [48]. This preprint was posted 26 August 2022, paper [20] was submitted 8 August 2022.
AZ: This sentence was added in the previous revision in response to the reviewer's assertion of priority of Aksenova at al. The fact that their article was submitted prior to the publication of our preprint does not prove their priority, as their article was only available to reviewers until it was published online in Internet, but we were not reviewers (this could be checked with the editorial office). In the present revision, the sentence in question has been removed, as the two studies were conducted with different methods, and the timing of their publication is of secondary importance.
P.S. E-mail to the Chinese (Cui et al.) authors:
Date: 25.07.2023, 15:57
From: Svetlana Hristova
To: <catcao1991@ibp.ac.cn>
Subject: Structural and functional characterizations of infectivity and immune evasion of SARS-CoV-2 Omicron
Hello,
Could you please write me at what pH the surface plasmon resonance was done in your research in the article: Structural and functional characterizations of infectivity and immune evasion of SARS-CoV-2 Omicron?
Dr. Hristova
Bulgaria
